# Bioactive Synthetic Polymer-Based Polyelectrolyte LbL Coating Assembly on Surface Treated AZ31-Mg Alloys

**DOI:** 10.3390/jfb14020075

**Published:** 2023-01-29

**Authors:** Sangeetha Kunjukunju, Abhijit Roy, John Ohodnicki, Boeun Lee, Joe E. Candiello, Mitali Patil, Prashant N. Kumta

**Affiliations:** 1Department of Bioengineering, University of Pittsburgh, Pittsburgh, PA 15261, USA; 2Department of Chemical and Petroleum Engineering, University of Pittsburgh, Pittsburgh, PA 15261, USA; 3Mechanical Engineering and Materials Science, University of Pittsburgh, Pittsburgh, PA 15261, USA; 4Center for Craniofacial Regeneration, University of Pittsburgh, Pittsburgh, PA 15261, USA; 5Center for Complex Engineered Multifunctional Materials (CCEMM), University of Pittsburgh, Pittsburgh, PA 15261, USA; 6McGowan Institute of Regenerative Medicine, University of Pittsburgh, Pittsburgh, PA 15219, USA

**Keywords:** MgAZ31, pretreatment layers, LbL coating, corrosion, BSA release

## Abstract

Polyelectrolyte layer-by-layer (LbL) films on pretreated Mg containing 3 wt.% Al and 1 wt.% Zn (MgAZ31) alloy surfaces were prepared under physiological conditions offering improved bioresponse and corrosive protection. Pretreatments of the model MgAZ31 substrate surfaces were performed by alkaline and fluoride coating methods. The anti-corrosion and cytocompatibility behavior of pretreated substrates were evaluated. The LbL film assembly consisted of an initial layer of polyethyleneimine (PEI), followed by alternate layers of poly (lactic-co-glycolic acid) (PLGA) and poly (allylamine hydrochloride) (PAH), which self-arrange via electrostatic interactions on the pretreated MgAZ31 alloy substrate surface. The physicochemical characterization, surface morphologies, and microstructures of the LbL films were investigated using Fourier-transformed infrared spectroscopy (FTIR), atomic force microscopy (AFM), X-ray diffraction (XRD), and scanning electron microscopy (SEM). The in vitro stability studies related to the LbL coatings confirmed that the surface treatments are imperative to achieve the lasting stability of PLGA/PAH layers. Electrochemical impedance spectroscopy measurements demonstrated that pretreated and LbL multilayered coated substrates enhanced the corrosion resistance of the bare MgAZ31 alloy. Cytocompatibility studies using human mesenchymal stem cells seeded directly over the substrates showed that the pretreated and LbL-generated surfaces were more cytocompatible, displaying reduced cytotoxicity than the bare MgAZ31. The release of bovine serum albumin protein from the LbL films was also studied. The initial data presented cooperatively demonstrate the promise of creating LbL layers on Mg-related bioresorbable scaffolds to obtain improved surface bio-related activity.

## 1. Introduction

Magnesium (Mg) and related alloys have been widely studied for their promise as bioresorbable scaffold materials for various biomedical technologies. Mg alloys provide high specific strength and desirable biocompatibility. Furthermore, the lightweight Mg and Mg-related alloys are more closely aligned with human bone [1,2,3]. Unfortunately, the principal issue with Mg-related biological structures is their rapid corrosive response in vivo, resulting in gas pockets due to accelerated degradation, causing undesirable accumulation of hydrogen gas. The rapid degradation of Mg-based implants can cause a hostile response to their neighboring settings and surrounding tissue with confined hydrogen gas pocket formation, causing a corresponding concomitant increase in pH and alkalization of the environment around the implant site, which have deleterious effects on the integration of the implant with the surrounding tissues [4]. To achieve a more desirable performance of Mg-based implants for biomedical applications, special surface modifications are used to decrease the rapid Mg implant degradation rate. Alterations of the surface of Mg and Mg-related alloys are aimed at creating an anti-corrosive layer with suitable surface biocompatibility. Correspondingly, developing processes that result in modifications of the surface have been observed to be an efficient method for improving the corrosive response and conferring abrasion-resistant characteristics to metal-related structures without altering the underlaying properties of the carrier construct and surface [5]. Myriad types of surface modification processes are recognized to date for improving the bioactivity and corrosion resistance properties of Mg-related alloys. These include chemically converted coating, spraying, electrochemical, or electroless film generation, organic coating, and laser or ion beam cladding [6,7,8]. Amongst the several corrosion protective film generation processes, chemically convergent surface treatment is largely the easiest, most successful, and by and large, unanimously characterized as a low-cost process. Various coating methods, including inorganic films, polymer depositions, and composite layers, are deployed after an initial pretreatment coating created on the Mg alloys. 

Polymer layers are often utilized to alter the properties of the surface of biomedical structures to enhance their bio-related compatibility, operation, and result in successful outcomes related to a particular treatment or therapy. Until now, polymeric layers are created by the layer-by-layer (LbL) self-arranged methodology. The LbL coating technique is prevalent for generating polymer films on Mg-related alloys since it is well-known and universally accepted that the approach is economical, benign for the environment, and also simple to perform [9]. Furthermore, the LbL coating method can produce polymer-related films with specific properties, such as bio-related compatibility, bio-related activity, and additionally even bacteriocidic characteristics. Additionally, the LbL arrangement approach is a flexible methodology implemented in drug delivery systems for localized applications and bio-related active coatings for bactericidal use [10,11]. Consequently, the LbL technique has been applied to a large range of biomedical systems, such as polyelectrolytes, nanostructured particles, polypeptides, DNA, proteins, enzymes, and dendrimers [12,13,14,15,16,17]. 

Entrapment of different molecules with precision and control on the layered surface helps to improve the bio-related activity and bio-related compatibility of these films, as well as facilitate fast restoration of the needed tissues. Exploiting these benefits attributed to LbL, various attempts to date have reported the use of this flexible approach to bestow corrosive protection as well as create bio-related active layers on Mg-related alloys [18,19,20,21,22,23]. Amongst the various polymers of natural and synthetic origin that have been studied, poly(lactide-co-glycolide) (PLGA) is ubiquitously used as a coating for the following reasons: 1. High biocompatibility; 2. FDA approved status; 3. Highly tailorable in vivo degradation rate, as well as drug delivery capability, and finally, 4. The degradation products of PLGA are metabolically digestible [24,25,26]. Therefore, the incorporation of PLGA over the Mg-related alloy surfaces could possibly act to provide corrosive protection and further help act as a temporary construct to release and deliver various antibiotic molecules. These include cefoxitin sodium, as well as common growth factors, namely, bone morphogenetic protein 2 (BMP-2). For biomedical applications, the desired drugs as well as signaling molecules are loaded into the implant materials, and consequently, achieving controlled and prolonged delivery can help in repairing, as well as restoring, the tissue. The controlled release of different biomolecules from LbL-coated Mg alloys has been reported in recent years. To date, several Mg-based drug-releasing orthopedic implant materials have been reported that focus on release of antibiotics, such as ciprofloxacin, vancomycin, and growth factors, such as BMP-2 [22,27,28]. 

The present study is focused on implementing the combination of an initial MgAZ31 pretreatment, consisting of magnesium hydroxide, Mg(OH)_2_ and MgF_2_ serving as a protective passive coating created by an in-situ chemically converted reaction. This is then followed by the creation of a layer-by-layer (LbL) coating of biodegradable polyelectrolyte multilayers conferring protection while also serving as a biologically compatible film surface created on AZ31 Mg alloy. The AZ31 Mg alloy was procured from a commercial vendor [18,19]. These LbL films created were characterized using morphological, physicochemical, and electrochemical techniques to assess the surface augmentation and thereby study the combined influence on the corrosive protection and resorption response of the AZ31 alloy beneath the film surface. Moreover, the compatibility of the layer-by-layer film generated on the AZ31 Mg surfaces toward the cellular response of the substrates was evaluated in vitro, by utilizing human mesenchymal stem cells (hMSCs). Since the LBL coating of the polyelectrolytes has a major influence on the release characteristics of biological molecules, this study also aims to incorporate protein into the LbL coating. Thus, the commonly used protein of bovine serum albumin (BSA) was used as a model protein to investigate the BSA protein release characteristics from the self-assembled, LbL films created on the AZ31 Mg alloy in order to demonstrate the protein release effectivity of the self-assembled polyelectrolyte layers. 

## 2. Experimental 

### 2.1. Preparation of Substrates

The MgAZ31 alloy in hot rolled form obtained from Alfa Aesar (Ward Hill, MA, USA) was utilized without any subsequent temperature or mechanical-cold work treatment. The commercially obtained AZ31 substrates were machined into 1.25 cm × 1.25 cm × 0.08 cm sizes and then subsequently cleansed by using an etchant of a solution of 3% nitric acid prepared in ethanol. The substrates following this etching treatment were subjected to sequential polishing utilizing 320, 600, and 1200 grit-sized papers of SiC, after which the substrates were carefully cleansed by ultrasonic washing utilizing acetone for 30 min. The cleaned AZ31 alloy substrates were then stored in desiccators until further use. 

### 2.2. NaOH and HF Pretreatment of Substrates

Polished alloys of AZ31 were immersed in 5 M NaOH solution at a temperature of 60 °C for a time-period of 3 h, after which they were cleansed completely utilizing deionized water, following which they were then kept for drying at 60 °C for 1 h. The fluoride coatings on hydroxyl-treated substrates were generated by immersion of the alloys following hydroxyl treatment in HF (48–51 wt.%, Acros Organics, NJ, USA) at ambient temperature for a period of 24 h with consistent stirring. The fluoride- and hydroxide-treated substrates were first subjected to a thorough washing step utilizing distilled water, after which they were rinsed with acetone and subsequently air-dried at 50 °C for 20 min, after which they were stored in a desiccator until further usage or testing. The hydroxide- and fluoride-treated substrates are denoted as FHAZ31. 

### 2.3. Generation of PLGA/PAH Multilayered LbL Layers

The surface modification scheme for the AZ31 alloys is shown in Figure 1. The LbL films were designed utilizing a sequence depicted as follows: “ABCBCBCBCB”. Solution A contained the cation containing polymer of polyethyleneimine (PEI, branched, Mw 70–150 kDa) at 2.5 mg/mL in DI water. The next layer was a solution of B that consisted of a synthetic anion containing polymer, 75:25 poly(lactide-co-glycolide) (PLGA 75:25, MW 76,000–115,000) at 50 mg/mL in dichloromethane. Solution C contained the cationic polymer poly (allylamine hydrochloride) (PAH, MW 58,000) used at a concentration of 5 mg/mL in 4-(2-hydroxyethyl) piperazine-1-ethanesulfonic acid buffer (HEPES buffer). The entire cationic polymer solution was rendered sterile by filtering using 0.22 µm filters prior to further usage. All the three polymers were purchased from Sigma-Aldrich (St. Louis, MI, USA). The FHAZ31-coated and bare AZ31 substrates were first immersed in solution A for a time-period of 2 min to create an initial coating that contained a fixed positive charge to commence the LbL self-arranging protocol. Alternate generation of the layers of the polyelectrolyte LbL film with subsequent rinsing was accomplished utilizing a dip coater (Desktop Dip Coater, MTI Corporation, Richmond, CA, USA, Model No EQ-HWTL-01-A). Following 2 min of immersion, the substrates coated with PEI were raised at a speed of 200 µm·s^−1^. Multiple layers of the polyelectrolytes were generated by sequential dipping of the substrates into the respective polymer solution and then, consequently, rinsed in deionized water for a time period of 1 min. Ultimately, 10 pairs of LbL coatings of multiple polyelectrolytes of (PLGA/PAH)10 that were coated with PLGA as the terminal layer were obtained. After that, all the LbL-generated samples were subjected to air drying at ambient temperature. The LbL-generated FHAZ31 substrates are referred to as FHAZ31L throughout the manuscript hereon. As mentioned above, bare, untreated AZ31 substrates were also coated with identical sequences of LbL coatings.

### 2.4. Evaluation of LbL Film Characteristics

The bare and non-treated, pretreated, and layer-by-layer coated substrates were examined using X-ray diffraction of glancing angle (GA-XRD, Philips PWI830 Diffractometer, Malvern Panalytical Ltd, Malvern, UK) utilizing copper Kα radiation with a 2θ range of 10–80° combined with 0.02° step size and using a ω angle of 1° offset. The chemical origin of the multilayered films was validated by attenuated total reflectance Fourier transform infrared spectroscopy (ATR-FTIR, Nicolet 6700 spectrophotometer, Thermo Electron Corporation, Waltham, MA, USA) utilizing a diamond ATR Smart orbit. The spectra were obtained in the 4000–500 cm^−1^ range. The morphology of the film surface was analyzed by scanning electron microscopy (SEM, JEOL JSM-6610LV) using a 10 kV operating voltage. Before imaging was performed, all the samples were subjected to coating by sputtering using a palladium (Cressington sputter coater 108A, Cressington, Watford, UK). The topography and roughness of the surfaces of the untreated AZ31, FHAZ31, and FHAZ31L were analyzed by non-contact oscillating atomic force microscopy (AFM, MFP3D Asylum Research, Oxford Instruments NanoAnalysis & Asylum Research, High Wycombe, UK). AFM was conducted at ambient temperature in air using a scan speed of 1 Hz at a resolution of 512 × 512 pixels using a silicon nitride conical tip with the characteristics of *k* = 40 N·m^−1^ (Mikromasch, Ltd, Tallinn, Estonia). Root mean square (RMS) surface roughness analysis was also conducted.

### 2.5. Coating Cytocompatibility Assessments 

#### 2.5.1. Assay to Assess Live and Dead Cells

The hMSCs that were isolated from human bone marrow were utilized in all the cell-based in-vitro analyses here (Lonza, Allendale, NJ, USA). After collection, all hMSC cells were passaged two times prior to cytocompatibility testing. The hMSCs were subjected to culturing at 37 °C in 5% CO_2_ and a humidity of 97%. The media used for culturing comprised minimum essential media α (α-MEM, Gibco, Grand Island, NY, USA) containing 20% fetal bovine serum (FBS, Atlanta Biologicals, Lawrenceville, GA, USA) and 1% penicillin streptomycin (P/S, Gibco, Grand Island, NY, USA). The coated and uncoated substrates were rendered sterile by subjecting to ultraviolet (UV) light exposure for 1h on each side. The sterilized substrates were placed in 12-well plates. In all the experiments performed, the cells were subjected to direct seeding on sterilized substrates, implementing a density of seeding of 30,000 cells per well. At day 1, as well as day 3, of cell culture, the samples were subjected to washing with phosphate buffer solution (PBS; obtained from Lonza Bio Whittaker, Walkersville, MD, USA, 1X, 0.0067 M (PO_4_) devoid of calcium or magnesium) and were then subjected to incubation for 30 min with calcein AM and ethidium homodimer-1 diluted in PBS (Invitrogen, Live/dead Staining Kit) at ambient temperature. Following incubation, all the samples were then again rinsed lightly using PBS before confining to fluorescence microscopy (Olympus CKX41, Olympus-Life Science, Waltham, MA, USA) imaging.

#### 2.5.2. DNA Quantification

The DNA concentrations on AZ31, FHAZ31, and FHAZ31L, and titanium (+control) were measured following the Quant-iT Pico Green dsDNA Kit (Invitrogen, Carlsbad, CA, USA). Accordingly, the cells were seeded directly onto the surfaces of the substrates at a cell density of 30,000 cells mL^−1^ and accordingly, cultured for 1, 3, and 5 days. At days 1, 3, and 5, the culture media was correspondingly removed, and all the samples were washed using PBS. Following this step, the cells were consequently lysed (CelLytic M Cell Lysis Reagent, Sigma Aldrich, St. Louis, MO, USA) and the respective supernatants were then collected following centrifugation. The DNA concentration in the supernatant was then assessed by determining the fluorescence using the wavelength for excitation of 480 nm and the corresponding wavelength of emission of 520 nm (Synergy 2 Multi-Mode Microplate Reader, Biotek). A sample size of n = 3 was used, and values were reported as average ± standard deviation. Statistical differences within the groups at each time point were then determined by performing a one-way ANOVA and Tukey post-hoc tests. For statistical analysis, a *p*-value of α < 0.05 was used to assess any statistical variabilities within the groups. 

### 2.6. Assessment of Stability of Films in Medium

The stability of the LbL films was verified by immersing the coated substrates in the Dulbecco’s modified Eagle’s medium (DMEM) containing 10% FBS and 1% P/S for 2 weeks and, correspondingly, refreshing the medium for every 24 h. Correspondingly, at different time periods, following immersion, any variations in the morphology of the film surface and any changes in the chemical functionality of the coated substrates were assessed utilizing SEM, as well as FTIR analyses, respectively.

### 2.7. In Vitro Assessment of Biodegradation and Corrosive Response

#### 2.7.1. Immersion Experiment and pH Changes

The in vitro biodegradation response of the untreated-bare, pretreated, and the LbL-deposited substrates were analyzed in hanks balanced salt solution (HBSS; obtained from Sigma Aldrich, St. Louis, MI, USA, H-1387). The deposited substrates were accordingly sterilized by subjecting them to UV light exposure for a time-period of a minimum duration of 30 min per side prior to immersion in HBSS. To assess the pH and concentration of the Mg-ion, each substrate with the film coating was placed inside 12-well tissue culture plates and fully submerged in 2.0 mL of HBSS. The submerged samples were maintained in an environment consisting of 37 °C, 5% CO_2_ as well as 95% relative humidity for a duration of 21 days. The HBSS was correspondingly altered and exchanged with freshly prepared HBSS after every 24 h period to assess the biodegradation rate in the respective in vitro environments. Three samples were correspondingly utilized for every film group (AZ31, FHAZ31, and FHAZ31L). The pH of the isolated HBSS was determined (350, Beckman Coulter, Brea, CA, USA) and kept at 4 °C for further analysis of the Mg ion concentration. The extracted HBSS was then subjected to dilution using 0.03 M Tris buffer (obtained from Sigma Aldrich), and the concentration of Mg-ion was analyzed utilizing inductively coupled plasma optical emission spectroscopy (ICP-OES; iCAP 6000 series, Thermo Electron Corporation, Waltham, MA. USA). Each ICP sample was determined using a minimum of three readings and reported as the average ± standard deviation. 

#### 2.7.2. Analysis of Electrochemical Response

Studies of the electrochemical response were determined using a CHI 604A (CH Instruments, Inc., Austin, TX, USA) workstation. Following the sample preparation protocol outlined earlier, the length and width of each sample were determined, and then one side of each test substrate was connected to a wire utilizing silver epoxy. After drying for 24 h, the epoxy of silver was then electrically insulated, implementing a resin of polymeric epoxy, ensuring that only each side of the sample was exposed for electrochemical testing. The test was conducted using a three-neck jacketed flask (ACE Glassware) filled with 125 mL DMEM containing 10% FBS and 1% P/S, that was maintained at equilibrium at 37 °C. All the test samples were allowed to reach equilibrium and attain a steady open circuit potential (OCP) prior to commencing the potentiostatic polarization analysis using a scan rate of 1.0 mV·s^−1^. The corrosive current and potential of corrosion were then determined using the Tafel extrapolation of the respective polarization curves. The corrosion current was then subjected to normalization by the exposed surface area to provide a measured value of the corrosion current density. The extrapolation of the Tafel curve was considered at the values of potential of 50 mV below and over the values of potential of corrosion to guarantee the correct extrapolated value. Origin software utilizing the Tafel extrapolation packet was implemented to perform the Tafel analysis and map the corresponding data points. Electrochemical impedance spectroscopy (EIS) analyses were conducted at open circuit potential for 10 mV sinusoidal amplitude using a frequency range of 100 kHz to 0.01 Hz. A steady OCP was maintained before commencing the tests. The OCP values obtained for bare AZ31, FHAZ31, and FHAZ31L were −1.61 V, −1.40 V, and −1.58 V, respectively. 

### 2.8. Loading and Release of BSA

BSA-loaded multilayered films were fabricated by LbL assembly based on the location of negatively charged polymers. The BSA (obtained from Sigma Aldrich; 98%, ~66 kDa) concentration of 2.5 mg/mL was incorporated after every two pairs of PLGA/PAH LBL films up to 20 layers. For the delivery of BSA, the multilayered film samples loaded with BSA were submerged in PBS (pH = 7.4). The BSA was also loaded on cleaned and non-pretreated LbL-coated bare AZ31, which was used as the control for comparing the BSA release characteristics of the LbL coatings on pretreated and bare AZ31 substrates. The released media were collected at chosen intervals of time and then restored utilizing a freshly prepared medium of identical volume. The concentrations of BSA in the corresponding solution were assessed utilizing the bicinchoninic acid (BCA) assay kit (Pierce Biotechnology, Waltham, MA, USA, Rockford Ill.). 

## 3. Results and Discussion

Surface modification is a widely used technique to counteract the undesirable rapid biodegradation rate of magnesium (Mg) and Mg alloys. Surface alteration, particularly the generation of layers of protection on the Mg alloys, has been considered an efficacious approach to decrease the early in vivo biodegradation rates of Mg alloys without adversely influencing the mechanical properties. Various research studies have been conducted thus far on Mg and Mg-related alloys for controlling the corrosion rate and enhancing bioactivity so that they can be used as a bioresorbable scaffold material [3,6,29]. To enhance the corrosive resistance and bioactive response of Mg and Mg alloys, the generation of surface modification coatings is essential. Consequently, the selection of the appropriate pre-coating surface treatment process is crucial for yielding the desired outcome of the coating on the Mg and Mg alloy response particularly with regards to cell and tissue response. A successful outcome for the coated surface requires that the non-toxic pre-coating surface treatments exhibit strong adhesion to the Mg substrate, thus providing the metal with the required protection from rapid corrosion and, correspondingly, improving the desired biocompatibility and/or bioactivity. Furthermore, considerable precaution is required to be considered during the entire process of deposition of the layers to combat the high chemical reactivity of Mg. Considering these requirements and to secure a desired positive outcome, a two-step treatment of the substrate surface utilizing hydroxide as well as fluoride interlayers was selected to enhance the corrosion resistance of the AZ31 alloy [30,31]. When assembling the polyelectrolyte multilayer films on the Mg alloy utilizing the LbL technique, it is desirable that the surface of the substrate possess a net charge of negative value. Thus, putatively, the selected alkali and hydrofluoric acid exposure on AZ31 alloys will result in conferring a negative charge to the surface due to the presence of surplus negatively charged hydroxide species and fluorine ion groups. Hence, these pretreated substrates will not only offer benefits during the arrangement of multilayered polyelectrolyte films, but the layers of oxide and fluoride species will also offer dual protection, by serving to doubly enhance the diffusion barrier for corrosive attack of hydroxyl and chloride ions in the implanted tissue and cellular environment, thus significantly reducing corrosion. In our previous work, we successfully constructed LbL coatings on AZ31 substrates that were initially treated with the hydroxyl ion and fluorine ions individually, as well as studied the effect on corrosion resistance, bioactivity, and cytocompatibility [18,19]. Thus, following this previous work, the current research is aimed at studying the effect of double pretreatments (hydroxide and fluoride) on the same substrates and further testing the introduction of an additional LbL coating serving as a delivery platform for controlled release of BSA biomolecules. In this work, all the polyelectrolytes and the corresponding organic polymers utilized to create the multilayers under room temperature conditions were biologically compatible and not toxic. Hence, in the current study, the LbL film generation process implemented will facilitate the incorporation of vital biological molecules into the structures containing multiple layers without affecting the stable nature of the individual as well as the assembled LbL-coated multilayers. 

### 3.1. Coating Characterization

Initially, XRD was utilized to validate the formation of hydroxide and fluoride layers on the AZ31 substrate surfaces. The AZ31 substrates were rinsed in distilled water after sodium hydroxide and hydrofluoric acid treatments. Figure 2 shows the collected XRD spectra for the AZ31, FHAZ31, and FHAZ31L substrates. The collected spectra show only major peaks of Mg, particularly at ~35.5° 2θ angle. Both the pretreated and layer-by-layer coatings generated are very thin; hence, it was not possible to detect any characteristic peaks corresponding to the pretreated and LbL-coated layers on the substrates. A dense inter-layer of MgF_2_ was however, formed on the alkaline treated AZ31 substrates, as shown in the SEM image (see Figure 3b). The hydroxide and fluorine ion conversion processes of coating are an easy and an efficacious method for ensuring the protection of the Mg and Mg-related alloy substrates from undergoing corrosion, as also documented by several reports in the literature [31,32,33]. 

FT-IR was utilized to analyze the chemical species in the layer-by-layer (LbL) generated films on the AZ31 alloy substrates. Figure 4 displays the FT-IR spectrum collected corresponding to the PEI, PAH, and PLGA polyelectrolytes used for the LbL coatings as well as the LbL-coated, pretreated substrates, and bare AZ31. The spectrum for the bare untreated substrate exhibits no peaks, as one will expect from metallic species since there are no infrared active centers on the surface of the AZ31 metal. The hydroxyl-deformation mode of water is implied by a weak broad band in the spectral domain of 1630–1650 cm^−1^ for FHAZ31. A peak below 600 cm^−1^ corresponds to presence of magnesium oxides or magnesium fluorides [31]. All characteristic peaks of PAH-, PLGA-, and LbL-coated pretreated substrates were observed within the wavenumber range of 500–4000 cm^−1^. The FTIR spectra of PLGA exhibited characteristic bands at 1750 cm^−1^ and 1080 cm^−1^ corresponding to the carbonyl, C=O stretching, and corresponding C–O–C bonds, respectively. The corresponding PAH characteristic peaks at 1605 cm^−1^ and 1519 cm^−1^ wavenumbers are assigned to the presence of N-H bond shifted to 1700 cm^−1^ and 1800 cm^−1^, respectively, likely due to the film thickness after the generation of the multilayers on the AZ31 substrate. The FHAZ31L substrate spectra collected and shown are obtained after completion of the LbL film creation protocol. The final top coating of the anion-containing polymer PLGA masked most of the signature molecular species peaks characteristic of the corresponding cation-containing polymers of PEI and PAH. 

Scanning electron microscopy was utilized to assess the morphology of the substrates after deposition of the layers. Figure 3 shows the SEM images of the bare polished surface of AZ31 (a) [18], pretreated (b), and following LbL film deposition on the AZ31 alloy substrate (c). The bare, untreated AZ31 substrate displays a flat and non-perfect surface containing trenches arising from the use of different grit sized SiC paper for surface polishing. The sodium hydroxide- and fluoride-treated side of the substrate, on the other hand, as expected, exhibits slightly different morphology of the substrate following creation of the hydroxide and fluoride layers arising from the conversion reaction of NaOH and HF reaction with the underlying Mg surface. As a result, the residual imperfections from the grooves created from the polishing process are visibly diminished and covered by the new hydroxide and fluoride layers. The LbL-coated substrates demonstrated smoother and dense surfaces showing very marginal remnants of the grooved surface imperfections created from the polishing artifacts remaining. The SEM results thus confirmed the successful formation of polyelectrolyte self-assembled layers due to the pretreatments as well as the anion–cation containing layer-by-layer (LbL) assembled layers formed on the top of the AZ31 substrates. 

To study the roughness of the substrate surfaces and assess the generation of any islands owing to the interlayer connections of the polyelectrolytes, the morphologies of the surfaces of the multiple-layered film substrates were further analyzed utilizing AFM. The AFM topographic images of bare AZ31, FHAZ31, and FHAZ31L substrates are displayed in Figure 5a–c, respectively. The images depict considerable differences between the bare, pretreated, and LbL-generated AZ31 substrates. The surface roughness (RMS) estimates determined for bare AZ31 were 80.73 nm. Following the alkali and fluorine ion treatments prior to LbL deposition (Figure 5b), the surface roughness decreased to 42.197 nm. The surface roughness values were further reduced to 25.60 nm after the LbL coatings on the pretreated alloy surfaces (Figure 5c). The AFM results thus revealed that all these surface pretreatments and LbL coatings led to significant alterations in the surface morphologies and roughness of the LbL coatings of AZ31. The FTIR, SEM, and AFM results collectively indicate that the AZ31 substrates were successfully coated with the polyelectrolytes following the pretreatments, with various numbers of layers generated by the layer-by-layer (LbL) process created by the technique of dip-coating. It is also well documented that alterations in physical-chemical and surface morphology characteristics can indeed affect surface–cell inter-relations [14,34]. Thus, the cell-related compatibility of all these coated samples was analyzed accordingly. 

### 3.2. Assessment of the Cytocompatibility of LbL Coating

#### Assessment of Cell Adhesion and Proliferation

For applications related to the biomedical field, it is important that any protection layer created on the Mg metal exhibits no toxicity and furthermore, should not negatively alter the bioactivity and biological response when used as an implant in a relevant animal model and ultimately if used in a human patient. Furthermore, the ideal protective coating should further improve the bioactivity of the Mg device [35]. Cellular adhesion onto the surface of the scaffold is therefore very important for the progress, preservation, and regeneration of the surrounding tissues. Cellular adhesion onto the bare surface of the Mg metal samples is difficult due to the high corrosion rate of the metal surface, which results in a rapidly changing surface with the concomitant evolution of hydrogen gas often resulting in cells to be lifted off the surface. Thus, the biocompatibility of the surface of the Mg scaffold must be improved to facilitate increased cellular attachment, viability, proliferation, and differentiation. The purpose of the pretreated and layer-by-layer (LbL) coatings presented here is to improve the surface biocompatibility of MgAZ31, thus facilitating early cell-related adhesion, proliferation, and, ultimately, attachment with the surface, thereby showing promise for its ultimate use if it were to be used as an implant. To assess the cytocompatibility and cellular adhesion prowess of these LbL-generated films, hMSC cells were directly cultured and grown for up to 5 days on bare, untreated AZ31, hydroxide- and fluoride-treated FHAZ31, and LbL coated following hydroxide and fluoride treatments, namely, FHAZ31L scaffolds. After the time period of 1, 3, and 5 days of cell culture, a live/dead cell staining assay was performed, and the cells were imaged using fluorescence microscopy. The corresponding staining assay results for liver and dead cells are shown in Figure 6.

Figure 6a–c reveals that very few hMSCs are attached to the bare AZ31 scaffold even after 5 days of culture, as expected due to the well-known reactivity of Mg and release of hydrogen gas, as mentioned earlier. On the other hand, an increase in cellular attachment and proliferation of hMSCs was observed in the hydroxide and fluoride pretreated samples compared to the bare AZ31 (Figure 6d–f). Figure 6g–i reveals that hMSCs cultured on the LbL-coated samples (FHAZ31L) demonstrated the most cellular attachment with elongated morphology after 1, 3, and 5 days of culture. Furthermore, what is interesting and worth noting is that after 5 days of culture, the surface of the FHAZ31L scaffolds was covered with a thick layer of hMSCs validating the beneficial in-vitro aspects of the pre-treatment and the subsequent creation of the LbL coatings on the AZ31 substrates. Dead cells were only present on top of the bare AZ31 samples at each time point which is not unusual and to be expected. The hMSC cell cultures with staining of live and dead cells, therefore, indicate enhanced viability and proliferation of the cells on LbL-generated film substrates contrasted with the uncoated pretreated AZ31 alloys as well as bare AZ31 alloys. Additionally, in comparison, only a few of the cells seen on the native bare AZ31 surface appeared rounded, indicating poor cellular attachment and cellular response. It should also be noted that much-reduced cell numbers were perceived on the bare Mg contrasted to the FHAZ31 samples after a time-period of 1, 3, and 5 days of culture. These cell adhesion and proliferation results further indicate that the pretreatment coating has a positive effect on rendering the Mg surface more biocompatible. Lastly, the final LbL coating produced the most biocompatible surface, resulting in the highest hMSC viability after 5 days of culture. In-depth cellular signaling analysis and molecular biology pathway analyses are, however, warranted in the future to gain more in-depth understanding of the cellular biology and interaction of the hMSCs with the bare AZ31, coated FHAZ31 and FHAZ31L surfaces. Even though the physicochemical performance of the LbL coatings could be the determining aspect for cell attachment important for dictating the application of these coated Mg alloys, ensuring cell proliferation over the long term is additionally dependent on other aspects. These include the LbL film stability over time and more importantly, assessing the local concentration variation of Mg ions, as discussed further in this manuscript. 

Figure 7 depicts the DNA concentration of hMSCs that were cultured on bare AZ31, pretreated FHAZ31, and LbL-coated FHAZ31L, as well as titanium foil serving as the positive control. The DNA content was determined by fluorescence assay. The amount of DNA determined is a direct account of the number of cells that attach to and proliferate on the substrate surfaces. The initial reduced DNA concentration and live cells attached to bare AZ31 indicate that many of the cells that were seeded did not attach in agreement with the live and dead cell stain assay results. This observation was anticipated for bare AZ31 owing to the known fast corrosion and hydrogen gas release characteristic of pure Mg and Mg alloys following the original early contact with the cell culture medium before the initiation of any protective layer. The initial cell attachment numbers on the FHAZ31 and FHAZ31L were statistically greater compared to the bare AZ31 at both days 1 and 3 of culture, as discussed earlier and shown in Figure 6. The DNA content of the LbL film-generated samples, FHAZ31L, was, however, statistically lower than FHAZ31, indicative of more cells being attached to the pre-coated surface in contrast with the live and dead cell staining assay. Although the live and dead cell staining seems to indicate the growth of a dense layer of hMSCs after day 5 of culture, the DNA quantification results are more representative of the total count of viable cells that are attached to the surface of the substrate. 

It is, nevertheless, vital to note that PLGA is largely understood to be semi-permeable to water, as the penetration and transport of water into the bulk of the polymer tends to be more rapid than the surface degradation via hydrolysis of the polymer [36]. The decrease in DNA concentration of the LbL substrates compared to the pretreated samples at each time point is therefore suspected to be due to the above behavior of PLGA, which likely allows the formation of pores and hence, limits cell growth on the entire surface of the LbL-coated substrates. Hence, the DNA concentrations measured on the LbL-coated surface were marginally lower than that of the pretreated surface. Furthermore, although the DNA concentration values for FHAZ31 and FHAZ31L are significantly lower than the bio-inert titanium control, the live/dead and DNA quantification results taken together demonstrate that both the pretreatment coatings and LbL coatings are effective at improving the cell-related compatibility of the bare AZ31 alloy surface, attesting to the efficacy of the use of these coatings strategies. As mentioned earlier, more in-depth cell-signaling pathways assessment using molecular biology techniques are needed in the future to gain a deeper understanding of these subtle variations. 

### 3.3. Assessing the In Vitro Stability of LbL Films in Medium

To assess the in vitro stability of the LbL films, these multiple layers coated substrates were submerged in DMEM for varying incubation times and assessed for changes in morphology and chemically functional molecular linkages utilizing SEM and FTIR. Figure 8a shows the FTIR spectra collected on FHAZ31L submerged and then subjected to aging in medium for 3, 7, and 14 days, respectively. These samples subjected to aging also displayed very similar FTIR spectra as that of the LbL-coated sample, as shown in Figure 4, with only a minor change observed in the positions of the peak. These results indicate that the coating is still being present on top of the surface of the FHAZ31L after 14 days following submersion. Figure 8b–d displays the SEM images (1000× magnification) corresponding to FHAZ31L subjected to aging for 3, 7, and 14 days. The images display a few small cracks being present and pores appearing on the surfaces of the films. The possible reason for these cracks in FHAZ31L is due to the surface hydrolytic nature of the polymer present in the top layer of the coating [36]. Furthermore, the pores and cracks seen on FHAZ31L in Figure 8b are sections where the hMSCs could not attach and proliferate and served to rationalize and justify the statistically decreased DNA concentration as compared to the pretreated, FHAZ31 surface after 3 days of culture. The corrosive resistance attribute perceived for FHAZ31 and FHAZ31L is most possibly owing to the underlying presence and creation of the MgF_2_ layer, which protects the surface from the rapid release of Mg^2+^ ions. The LbL films deposited on the fluorine ion-treated substrates appear not to be affected throughout the entire 7 days of the submersion test. To further study the role of resistance to the corrosive response conferred by the pretreatments and the presence of multilayer coatings, the release of Mg ions and the corresponding charge transfer resistance of these test systems were further determined. These results are described in the sections to follow below. 

### 3.4. Assessment of In Vitro Biodegradation 

A long duration submersion test is required to be conducted to estimate the biodegradation response of the coated samples, which will offer more convincing and practical information about the in vitro stable nature of the samples. The variation in Mg^2+^ ion amounts in the soaking medium of HBSS over different time periods during the entire non-static submersion tests are displayed in Figure 9. The inductively coupled plasma (ICP) optical emission spectrometry (ICP-OES) assessment showed a fast release of the doubly charged magnesium ion from the bare AZ31 samples, which was clearly not seen in either the FHAZ31 or FHA31L samples (Figure 9a). The pretreated FHAZ31 samples, however, displayed a prolonged release of dual-charged magnesium ions throughout the entire time period of the study. The amount of dual charged magnesium ions released from the FHAZ31, hydroxide and fluorine ion-coated substrate was much reduced than that released from bare AZ31. Similar response behaviors were also perceived for the LbL-deposited substrates following pretreatment. These observations thus, clearly reflect that all the hydroxide and fluorine ion-treated and magnesium fluoride-coated surfaces show more corrosione resistance and prevent the immersion medium from reacting with the AZ31 metal. Prior studies have also documented the generation of a biodegradation layer during the entire process of corrosive response of magnesium alloys, and it has been construed that this biodegradation layer itself lowers the corrosion rate [37]. Figure 9b displays the pH variation in the HBBS medium (initial pH = 7.3 at 37 °C) that contains the bare AZ31 substrate, pretreated, and LbL-deposited substrates. Results show that the values of pH of all the extracted test specimens were slightly in the basic pH region; however, the extracts from the pretreated substrates and LbL-deposited substrates were less basic in pH contrasted to bare and untreated AZ31 for time points below 14 days. Beyond 14 days, it appears that extracts from both pretreated and LbL-coated samples following pretreatments exhibited slightly higher alkalinity compared to the bare AZ31 samples. This slight variation could be assigned to the existence of porosity in the LbL-coated samples discussed earlier due to the PLGA layer and possible defects of cracks and pinhole pores present on the pretreated sample surface arising due to the grooves originally present on the surface of the bare AZ31 samples created from polishing with the SiC paper, as discussed earlier and seen in Figure 3.

### 3.5. Electrochemical Corrosion Characterization

The corrosion endurance of the bare and surface-altered and treated substrates was analyzed using potentiodynamic polarization (PDP) in DMEM medium. The PDP curve is an efficacious and popular method for assessing the corrosion endurance of biomaterials, particularly biodegradable metals. The Tafel curves offer valuable data on the corrosion potential value and the corresponding current density, as well as provide a representative pictural description of the corrosion response properties of the test samples at the time of estimation. In general, the more positive or higher the value of the electrochemical potential for corrosive response, which is more representative of the ensuing equilibrium corrosion reactions, and the reduced corrosion current density are both indicative of better corrosion resistance of the pretreated and LbL-coated samples following NaOH and HF pretreatments of AZ31. Figure 10a correspondingly displays the typical curves for the potentiodynamic polarization of the bare AZ31, FHAZ31, and FHAZ31L samples. The corrosion response potential (Ecorr) was determined to be −1.591 V, −1.392 V, and −1.496 V for AZ31, FHAZ31, and FHAZ31L, respectively. The corrosion current density (icorr) for AZ31, FHAZ31, and FHAZ31L was 7.97 × 10^−6^ A/cm^2^, 3.09 × 10^−7^ A/cm^2^, and 6.87 × 10^−7^ A/cm^2^, respectively. Thus, it can be concluded that both FHAZ31 and FHAZ31L exhibited increased corrosion resistance compared to bare AZ31. As expected, based on the above analysis, the pretreated FHAZ31 sample was the most corrosion resistant of all the AZ31 samples tested. It can be noticeably mentioned that the compact and dense form of magnesium hydroxide films and the MgF_2_ layers serve to protect the underlying substrate created by the sodium hydroxide and hydrofluoric acid reaction processes and can thus enhance the corrosive endurance of the Mg alloy, in line with published reports [30,38,39,40]. These results unequivocally demonstrate that the corrosion resistance of the AZ31 alloy surface was enhanced predominantly by the dual alkali and acid surface pretreatments. 

Unlike PDP, however, EIS is not a destructive technique and is characterized by frequency-related variations in complex impedance, which are measured in response to an alternating current probe voltage. Thus, the data connected to the corrosion protection and endurance of the test samples under evaluation can be procured, which is particularly beneficial when determining the capability of a deposited film to offer a protective response to the test sample underneath. The corrosion endurance of the films correspondingly in DMEM medium was analyzed utilizing EIS, and the experimental data obtained are displayed in Figure 10b–d. After the dual alkali and acid surface treatments and LbL coating, the semicircle diameters of the Nyquist plots are larger than those corresponding to the bare, untreated AZ31 alloy. Hydroxyl and fluorine ion treated and the corresponding chemically converted substrates show a rise in the diameter of the semi-circle with an intercept resistance on the Z’ axis of ~115,000 Ω, which is indicative of the resistance to transfer of charge that occurs at the alloy and the DMEM solution interface, contrasted to that of the bare, untreated substrate (~2500 Ω). These findings further support and attest to the corrosive endurance of the Mg (OH)_2_ and fluoride layers formed on the AZ31 alloy following the dual alkali and acid treatments. The LbL-coated substrates following the NaOH, and HF pretreatments also show an elevation in the semi-circle diameter with an intercept resistance on the Z’ axis of ~110,000 Ω as compared to bare AZ31. The creation of the oxide and fluoride layers following dual alkali and acid pre-treatments and addition of the LbL layers lowers the charge transfer resistance compared to FHAZ31 since LbL layers are typically porous, as explained earlier, and the impedance test data imply that the corrosive protection prowess originates mainly from the basic pH-related hydroxyl and acidic fluorine-ion pretreatments of the substrate. The marginal decrease in the charge-transfer resistance values of the FHAZ31L substrates owing to the porous PLGA layer does not however, diminish the propensity of the LbL coatings offering improved corrosion protection to the underlying AZ31 substrate.

Further analysis of the coatings was conducted by fitting the collected experimental EIS data of each material system to the equivalent circuit models shown in Figure 10e. The solid red line in Figure 10b–d represents the modeled Nyquist plots. For AZ31 and FHAZ31, two constant phase element (CPE) loops were observed in the equivalent circuit, showcasing that neither the Mg substrate nor the pretreatment layers or inherent oxide layer on the Mg exhibited the features of a perfect capacitor, thus implying the presence of a double layer. However, the addition of the polymeric LbL coating on alkali and HF pretreated AZ31, namely, HFAZ31L required the addition of a third loop consisting of an ideal capacitor in parallel with a resistor to adequately model the experimentally obtained EIS plot. The parameters obtained from the equivalent circuit modeling of each system, shown in Table 1, further illustrate the effectiveness of the pretreatment and addition of the LbL coating layers in increasing the overall charge transfer resistance of the bare Mg alloy in DMEM. The addition of the hydroxyl and fluorine ion-treated FHAZ31 substrate layer increases the resistance, R_coating_ = 113,000 Ω, in the equivalent circuit as compared to bare AZ31, R_oxide_ = 2233 Ω. On the other hand, in agreement with the experimentally obtained EIS results, the resistance of the FHAZ31L equivalent circuit model (R_polymer_ + R_coating_) is slightly reduced to 104,279 Ω after the addition of the LbL polymeric layer. The surface pretreatments thus, additionally inhibited the effective solubility of the underlying AZ31 substrate to a large extent, conferring long-term stability for the LbL deposited layers on the pretreated AZ31 alloys to offer improved bioactivity as well as cytocompatibility. Hence, the prospects of these LbL-coated alloys for protein release were further evaluated, as explained in the next section. 

### 3.6. BSA Release

Based on research conducted in the last few years, the release of biomolecules from magnesium alloys has gained noticeable importance and has also emerged as a research area of much interest in the biomedical applications arena. There are various documents published in the literary reports outlining the use of various biomolecules that have been incorporated into different coating systems on Mg-based alloys, with concomitant studies of the release profiles and the evaluation of the associated bioactivity of the system. 

Table 2 here provides an overview of the controlled release of different biomolecules incorporated into the various polymeric, inorganic, and composite coatings created by various methods on Mg and Mg alloys. Min-Ho Lee et al. studied the tethering of BMP-2 in LbL containment layers (micro-arc oxidation (MAO) layers combined with hydrothermal treated process) on magnesium alloy and determined that the film significantly improved bone generation propensity based on the amount of BMP-2 immobilized and released [22]. Combining the electrostatic interactions of polyelectrolytes with LbL self-assembly techniques, the efficacy of this coating as a delivery platform for the release of BSA as the model protein to assess the release profile over an extended time-period was tested. It is well known that BSA, a globular protein carrying a negative charge and possessing an average molecular weight of 66 kDa comprising 580 amino acid residues, is extensively used in controlled release studies. The BSA containing LbL coatings on pretreated and bare AZ31 substrates in the present work were subjected to incubation in PBS at 37 °C for 10 days, and the amount of BSA released was determined quantitatively by micro-BCA protein assay. Figure 11a,b show the release profiles of BSA from the LbL coated bare and LbL coated on pretreated substrates over a time-period of 10 days. As displayed in Figure 11, the release response of BSA from bare LbL-coated AZ31 and FHAZ31L was significantly different. Figure 11a profiles describe the influence of the pretreatment coating on the FHAZ31L sample, thereby facilitating a higher maximum BSA cumulative release (660.18 ± 3.11 µg/mL) over 10 days as compared to the LbL-coated bare AZ31 (128.99 ± 0.58 µg/mL). Thus, the FHAZ31L sample resulted in a 19.5% higher BSA release over 10 days compared to the LbL coatings on bare untreated AZ31. Furthermore, as displayed in Figure 11b, a 66% burst release of BSA was observed from the LbL-coated non-pretreated substrates within the initial first day. It is possible that BSA was rapidly released because it was loosely attached in the layers through purely ionic interactions, and that it could be easily detached from the ionic environment. The burst release of BSA from the LbL-coated bare substrates could also be caused by the inferior corrosion resistance of bare AZ31 in the medium. On the other hand, BSA incorporated into the multilayer LbL coatings on the pretreated substrates exhibited 44% initial bursts within day 1, followed by sustained release, reaching up to 93% after 7 days under physiological conditions. The sustained release of BSA from the LbL coated FHAZ31 substrates was observed to occur due to the better corrosion resistance obtained from the dual action of NaOH and HF pretreatments on the AZ31 substrate. Compared with non-pretreated substrates, BSA release from surface-treated substrates was more persistent. This surface treatment clearly facilitates the stability of the LbL films and offers more controlled release, as well as higher cumulative release of BSA. These initial results, therefore, clearly demonstrate the promising nature of the use of the LbL coatings on the pretreated AZ31 alloy, suggesting that this approach should be implemented as a preferred strategy for achieving controlled release of proteins and growth factors, which is the objective of the current work. The improved cell attachment observed on the FHAZ31L samples discussed earlier also serves as a testament to the use of this approach as a coating platform strategy for effectively delivering various drugs and proteins in a controlled fashion. Further studies are, however, clearly warranted to determine the influence of each LbL layer, including the thickness and concentrations needed to determine the optimal LbL coating parameters for achieving the desired optimal dosage of growth factors at a given time point with optimal cytocompatibility, which are part of the future work. Similarly, the long-term durability of the pre-treatments and the created LbL coatings on the pre-treated surface combined with the stability and shelf-life of the encapsulated growth factors need to be explored which are clearly pathways for subsequent follow-on work. The approach as demonstrated here using BSA as the model protein is indeed versatile and, when optimized, can be applicable and extended to any biodegradable Mg alloy that has been developed by us and several other researchers in the biodegradable metallic alloy field.

## 4. Conclusions

Multilayered polyelectrolyte films of PLGA and PAH were successfully synthesized using a straightforward LbL coating technique on AZ31 alloy substrates that were exposed to dual action hydroxyl and fluorine ion pretreatments. The double-layer pretreatment surface layer effectively lowered the corrosive response of Mg alloy, AZ31 and effectively reduced the burst release of BSA from the LbL coatings from 66% for the bare untreated AZ31 alloy to 44% for the pre-treated AZ31 alloy, a reduction of 22% while providing 19.5% higher cumulative release over 10 days. The LbL multilayers on the dual action pre-treated AZ31 alloy showed a sustained release profile of BSA for a period of 10 days. Furthermore, the additional LbL films created on the dual alkali and acid pretreated Mg alloy substrates also provided improved cytocompatibility and bioactivity, contrasted with that of the uncoated AZ31 alloy substrates. These results and the concept presented here may have great potential for biomedical applications, particularly those related to drug and protein delivery.

## Figures and Tables

**Figure 1 jfb-14-00075-f001:**
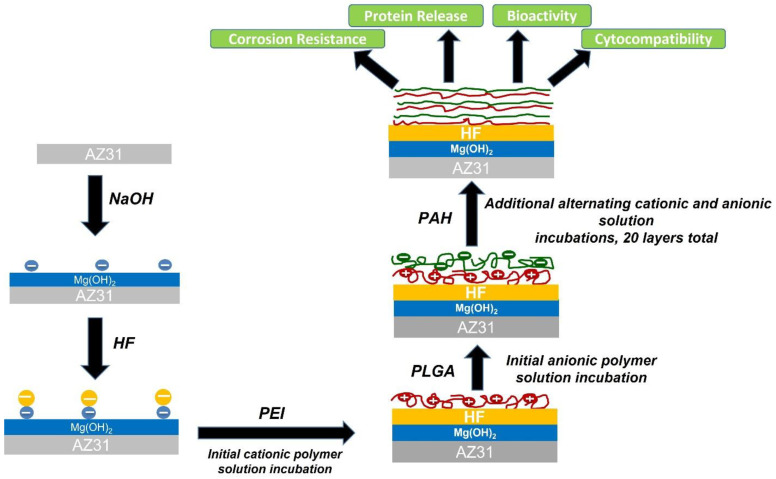
Schematic representation of pretreatments and LbL film preparation on bare AZ31 alloy substrate.

**Figure 2 jfb-14-00075-f002:**
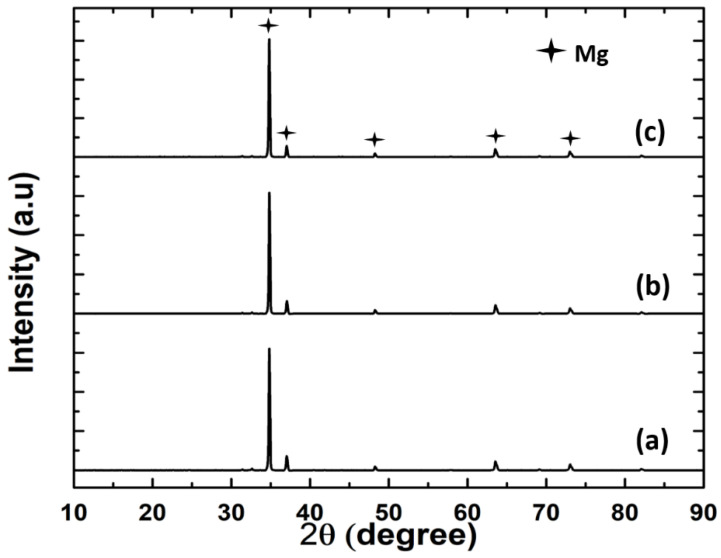
X-ray diffraction patterns of (**a**) bare AZ31, (**b**) FHAZ31, and (**c**) FHAZ31L coated AZ31 alloy substrates.

**Figure 3 jfb-14-00075-f003:**
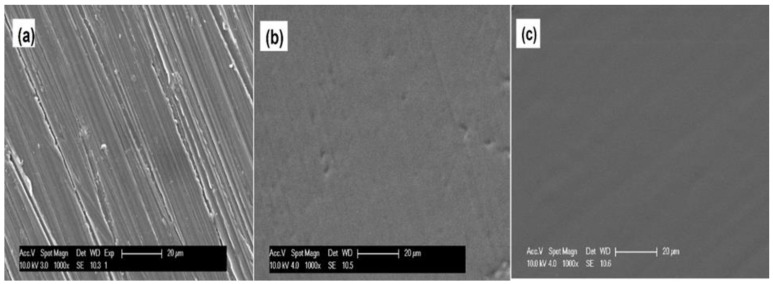
SEM images at 1000X magnification of (**a**) Bare AZ31 [18], (**b**) FHAZ31, and (**c**) FHAZ31L coated AZ31 alloy substrates.

**Figure 4 jfb-14-00075-f004:**
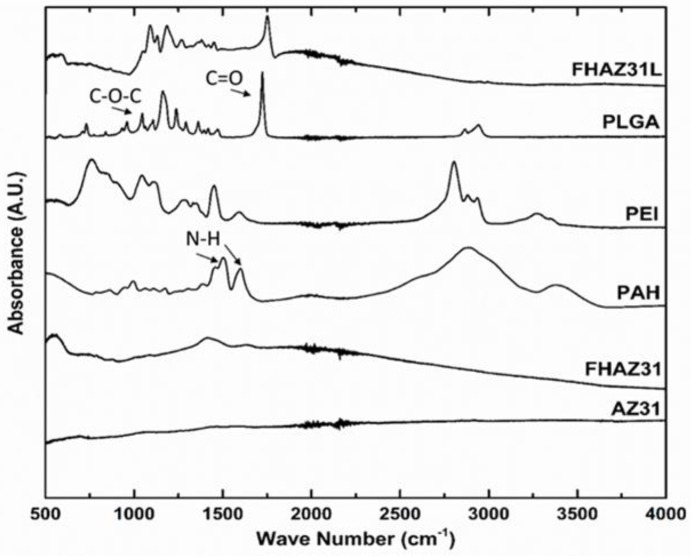
FTIR spectra of bare AZ31, FHAZ31, FHAZ31L coated substrates, PAH, PEI, and PLGA.

**Figure 5 jfb-14-00075-f005:**
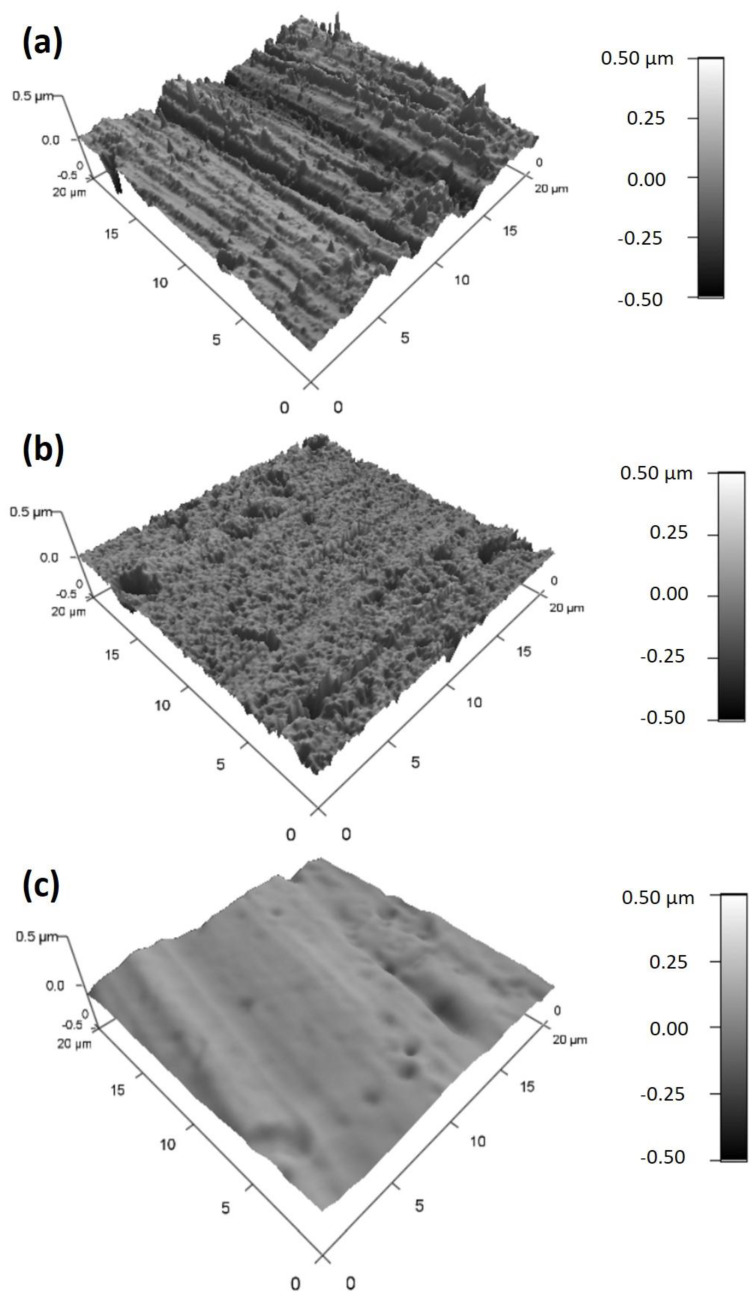
AFM images collected on (**a**) Bare AZ31, (**b**) FHAZ31, and (**c**) FHAZ31L coated AZ31 substrates.

**Figure 6 jfb-14-00075-f006:**
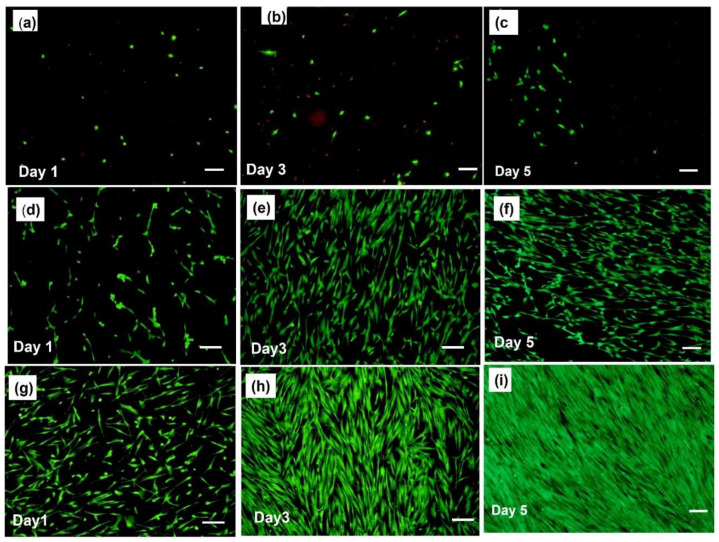
Microscopic images of live and dead cells attached on Bare AZ31 at days 1, 3, and 5 (**a**–**c**) FHAZ31 at days 1, 3, and 5 (**d**–**f**) FHAZ31L at days 1, 3, and 5 (**g**–**i**) (scale bar = 200 µm).

**Figure 7 jfb-14-00075-f007:**
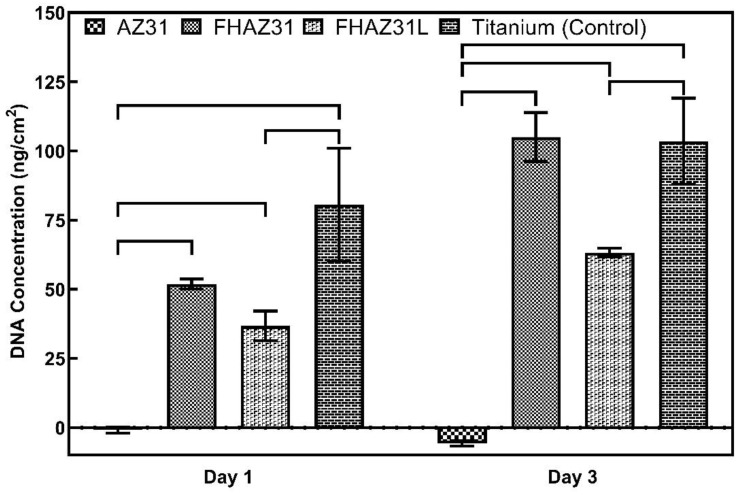
Quantification of total DNA content using hMSC cells on substrates at various time points. Statistical significance determined at *p* ≤ 0.05.

**Figure 8 jfb-14-00075-f008:**
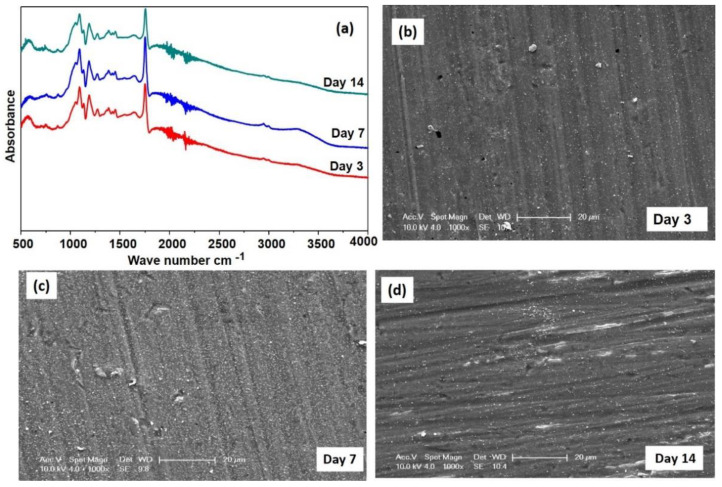
(**a**) FTIR spectra of FHAZ31L in medium for 3, 7, and 14 days; SEM images of FHAZ31L in medium for (**b**) 3 days, (**c**) 7 days, and (**d**) 14 days.

**Figure 9 jfb-14-00075-f009:**
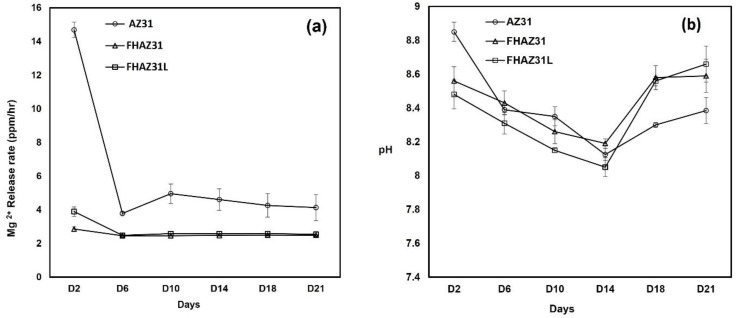
(**a**) Magnesium ion release using ICP-OES measurements of AZ31, FHAZ31, and FHAZ31L substrates. (**b**) pH profile of AZ31, FHAZ31, and FHAZ31L in HBBS medium over 21 days.

**Figure 10 jfb-14-00075-f010:**
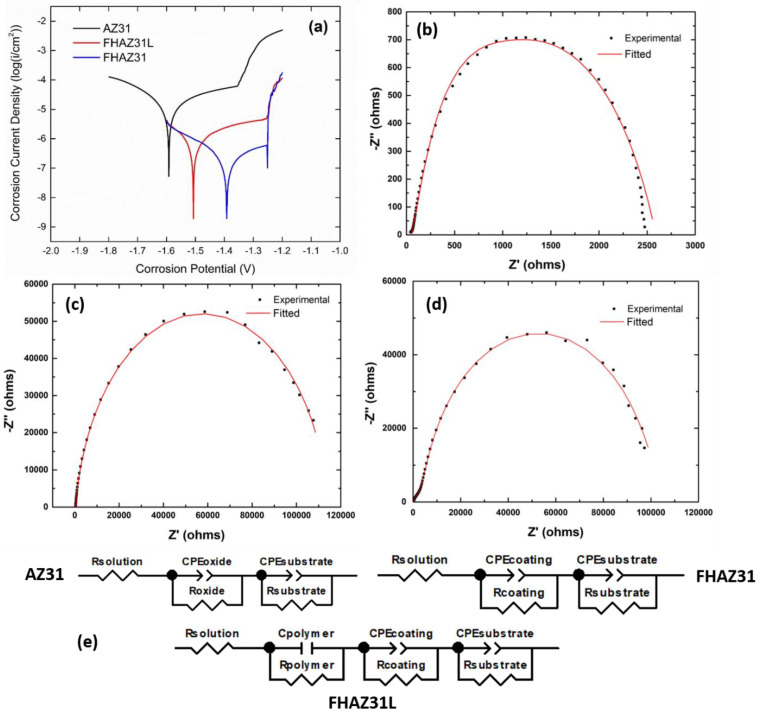
(**a**) Tafel plots of AZ31, FHAZ31, and FHAZ31L (**b**) Experimentally obtained and theoretically fitted Nyquist plots obtained on bare, AZ31 (**c**) Experimentally obtained and theoretically fitted Nyquist plots obtained on pretreated AZ31 alloy, FHAZ31 (**d**) Experimentally obtained and theoretically fitted Nyquist plots obtained on the pretreated and LbL-coated AZ31 alloy, FHAZ31L, and (**e**) representative equivalent circuit model for each of the material systems. (CPE = constant phase element, R = resistance, and C = capacitance).

**Figure 11 jfb-14-00075-f011:**
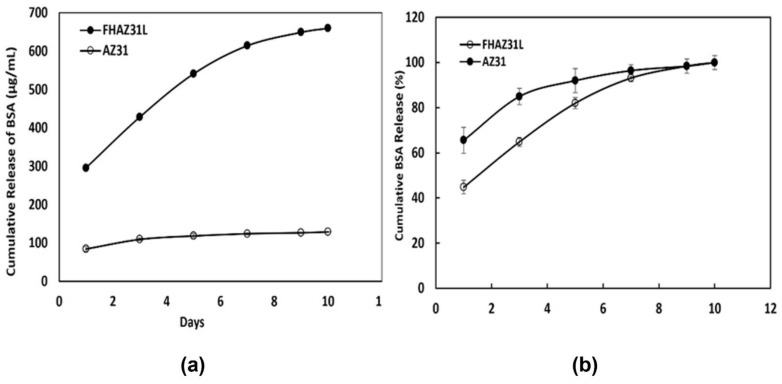
(**a**) Cumulative amount of BSA released from LbL coatings on bare AZ31 and LbL coatings on pretreated FHAZ31L; (**b**) Percentage cumulative release of BSA released from bare AZ31 and FHAZ31L.

**Table 1 jfb-14-00075-t001:** Equivalent circuit modeling parameters for AZ31, FHAZ31, and FHAZ31L.

Material Systems	AZ31	FHAZ31	FHAZ31L
R_solution_ (Ω)	51.25	441.6	57.35
R_oxide_ (Ω)	2233	-	-
R_coating_ (Ω)	-	1.13 × 10^5^	23,613
R_polymer_ (Ω)	-	-	80,666
R_substrate_ (Ω)	305.6	757.7	1951
CPE_oxide_-T (F)	7.60 × 10^−5^	-	-
CPE_oxide_-P	0.633	-	-
CPE_coating_-T (F)	-	1.98 × 10^−5^	7.61 × 10^−5^
CPE_coating_-P	-	0.9474	0.559
C_polymer_ (F)	-	-	2.19 × 10^−5^
CPE_substrate_-T (F)	2.38 × 10^−5^	0.00027	1.38 × 10^−5^
CPE_substrate_-P	1.135	0.99	0.792

**Table 2 jfb-14-00075-t002:** Summary of the controlled release studies of various biomolecules and the different coating methods employed to immobilize the biomolecules on Mg and Mg alloys.

Year	Coating Method	Substrate	Biomolecules	References	Remark
2016	Electrode Deposition	MgAZ31	Ibuprofen	J Solid State Electrochem (2016) 20:3375–3382 [37]	Rapid release of Ibuprofen appeared
2016	Composite film coating	MgAZ91	Ibuprofen	Materials Science and Engineering: C Volume 68, 1 November 2016, Pages 512–518 [41]	Initial rapid release and further sustained release of Ibuprofen were observed
2018	Polymer Spray coating	MgAZ31	Sirolimus	Colloids and Surfaces B: Biointerfaces Volume 163, 1 March 2018, Pages 100–106 [42]	Initial rapid release and then further stable release of Sirolimus was observed
2019	LbL coating	MgAZ31	Gentamicin (GS)	Journal of Colloid and Interface Science Volume 547, 1 July 2019, Pages 309–317 [43]	Prolonged release profile of GS was observed
2018	MAO coating	MgAZ31	Ferulic acid	Applied Surface Science Volume 435, 30 March 2018, Pages 320–328 [44]	Initial burst release of Ferulic acid appeared
2011	MAO coating	MgAZ81	Paclitaxel (PTX)	Colloids and Surfaces B: Biointerfaces Volume 83, Issue 1, 1 March 2011, Pages 23–28 [45]	Nearly linear sustained release of PTX with no significant burst releases and prolong the release time were observed
2021	LbL coating	MgAZ31	Ciprofloxacin	Applied Surface Science Volume 508, 1 April 2020, 145240 [46]	Rapid release of Ciprofloxacin was observed
2019	Hydroxyapatite coating	MgAZ31	Gentamicin sulfate	Front. Mater. Sci. 2019, 13(1): 87–98 [11]	Burst release of GS and then long-term slow release was observed
2019	Hydroxyapatite and LbL Composite coating	MgAZ31	Gentamicin	Colloids and Surfaces B: Biointerfaces Volume 179, 1 July 2019, Pages 429–436 [23]	Burst release of GS during the initial 24 h and then subsequent long-term slow release observed for 15 days

## Data Availability

The raw/processed data required to reproduce these findings cannot be shared at this time, as the data also forms part of an ongoing study. However, the authors will be glad to share the relevant information upon request.

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
