# Peer review of "Bioactive Synthetic Polymer-Based Polyelectrolyte LbL Coating Assembly on Surface Treated AZ31-Mg Alloys"

_jfb, 2023, doi:10.3390/jfb14020075_

Round 1

Reviewer 1 Report

132 - you should correct  2.5 mg ml/mL in DI water in 2.5 mg/mL in DI water

136-137 -you should correct 5 mg ml/mL in 5 mg/mL

310-316 - you affirmed “Fig. 2 shows the collected XRD spectra for AZ31, FHAZ31 and FHAZ31L substrates. The collected spectra show only major peaks of Mg, particularly at 35.5° 2θ angle. Both the pretreated and layer by layer coatings generated are very thin and hence, it was not possible to detect any characteristic peaks corresponding to the pre-treated and LbL coated layers on the substrates. A dense inter layer of MgF2 was however, formed on the alkaline treated AZ31 substrates, as shown later in the SEM image (see Figure. 4b)” - but in your publication N. Ostrowski, B. Lee, N. Enick, B. Carlson, S. Kunjukunju, A. Roy, P.N. Kumta, Acta Biomaterialia, 9 (2013) 8704-8713 you observed a difference using same NaOH treatment “NaOH treated AZ31 substrate showing growth of Mg(OH)2 layer, indicated by the asterisk” Figure 1 – how its possible?

“Following sodium hydroxide treatment and rinsing in DI water, the substrate was dried and kept in a desiccator until the diffraction measurement was completed. Fig. 1 shows the collected GA-XRD spectra along with simulated spectra for pure magnesium and magnesium hydroxide. The collected spectra show minor peaks associated with major peaks of Mg(OH)2 spectra, particularly at ~18.5° and 38° 2θ angles”

343 - Figure 4 shows SEM images of the bare polished surface of AZ31 (a), pretreated (b), and following LbL film deposition (c). – In Figure 4 you have exactly the same image of the bare polished surface of AZ31 (a) as the one the you first reported in 2013 Reference 18 Acta Biomaterialia, 9 (2013) 8690-8703 (Figure 1d), I have doubts about the fact that you used the same material, and if it is a standard image maybe you should quote it.

516 - Figure 9. Magnesium ion release using ICP measurement of AZ31, FHAZ31, and FHAZ31L (b) pH profile of AZ31, FHAZ31, and FHAZ31L in DMEM medium over 21 days.- you forgot to put (a) in front of magnesium ion release

Author Response

Response to Reviewer 1. All corrections are highlighted in yellow in the revised manuscript.

  1. 132 - you should correct 2.5 mg ml/mL in DI water in 2.5 mg/mL in DI water

Response: Thank you for pointing this out. The sentence in line 134 is corrected as 2.5 mg/mL in DI water highlighted in yellow.

  1. 136-137 -you should correct 5 mg ml/mL in 5 mg/mL

Response: Thank you for pointing this out. The sentence in line 137-138 is corrected as 5 mg/mL in 4-(2-hydroxyethyl) piperazine-1-ethanesulfonic acid buffer (HEPES buffer) highlighted in yellow.

  1. 3. 310-316 - you affirmed “Fig. 2 shows the collected XRD spectra for AZ31, FHAZ31 and FHAZ31L substrates. The collected spectra show only major peaks of Mg, particularly at ∼5° 2θ angle. Both the pretreated and layer by layer coatings generated are very thin and hence, it was not possible to detect any characteristic peaks corresponding to the pre-treated and LbL coated layers on the substrates. A dense inter layer of MgF2 was however, formed on the alkaline treated AZ31 substrates, as shown later in the SEM image (see Figure. 4b)” - but in your publication N. Ostrowski, B. Lee, N. Enick, B. Carlson, S. Kunjukunju, A. Roy, P.N. Kumta, Acta Biomaterialia, 9 (2013) 8704-8713 you observed a difference using same NaOH treatment “NaOH treated AZ31 substrate showing growth of Mg(OH)2 layer, indicated by the asterisk” Figure 1 – how it’s possible? “Following sodium hydroxide treatment and rinsing in DI water, the substrate was dried and kept in a desiccator until the diffraction measurement was completed. Fig. 1 shows the collected GA-XRD spectra along with simulated spectra for pure magnesium and magnesium hydroxide. The collected spectra show minor peaks associated with major peaks of Mg(OH)2 spectra, particularly at ~18.5° and 38° 2θ angles”

Response:  Prior work shows that Mg (OH)2 coatings on AZ31 (New_Reference_1; Tang et al. Int. J. Electrochem. Sci. 12 (2017) 1377-1388 see attached) can range from 30-70um based on immersion time, whereas HF coatings on Mg are between 1-3um (New_Reference2; da Conceicao et al. Thin Solid Films (2010) 5209-5218).  Thus, our prior work with Mg (OH)2 showing XRD peaks characteristics of Mg (OH)2 can be attributed to the presence of thick layers of the hydroxide.  However, HF coatings on Mg typically do not show any MgF2 characteristic peaks due to the much-reduced thickness.  In our data, the lack of Mg(OH)2 characteristic peaks indicates that the majority of the surface is covered by a thin coating of  MgF2

  1. 343 - Figure 4 shows SEM images of the bare polished surface of AZ31 (a), pretreated (b), and following LbL film deposition (c). – In Figure 4 you have exactly the same image of the bare polished surface of AZ31 (a) as the one the you first reported in 2013 Reference 18 Acta Biomaterialia, 9 (2013) 8690-8703 (Figure 1d), I have doubts about the fact that you used the same material, and if it is a standard image maybe you should quote it

Response:  Yes. We have used the same bare material. We have quoted the reference in the text in line 345 and in the figure caption of Figure 4 highlighted in yellow.

  1. 516 - Figure 9. Magnesium ion release using ICP measurement of AZ31, FHAZ31, and FHAZ31L (b) pH profile of AZ31, FHAZ31, and FHAZ31L in DMEM medium over 21 days.- you forgot to put (a) in front of magnesium ion release

Response: We have added (a) in the caption of Figure 9 highlighted in yellow.

Reviewer 2 Report

I have sent by mail 

Author Response

Thanks to the authors for their interesting work. I have some points about the manuscript to mention here.

  • The corrosion resistance of the coating system is very important as mentioned by the authors too, but is not elaborated enough. The water permeability of the LbL coating (PLGA) will deteriorate the system and generates hydrogen gas which is not mentioned in the manuscript about it.

Response: The focus of the work was more on exploring the influence of alkali and acid pretreatment on generation of LbL coatings and the feasibility of the composite coatings to bind and release BSA, a model protein over two weeks. Instilling any possible corrosion protection is a likely added benefit but not the primary focus. However, as part of the characterization completeness, we have conducted EIS measurements and have also modeled the data showing both the experimental and fitted plots. We have also analyzed the circuit parameters and have commented on the increase in the resistances due to the coatings. The release studies and the cytocompatibility of the coated films show that the film is stable for a period of 10 days while the ICP measurements also show no increase in Mg2+ ion release rate for 21 days. These results therefore confirm that the dual treatment and generation of the LbL layers make the system stable for the period of study in this work. The aim was not to assess the long-term stability of the system. This would be part of the future work that the authors will plan to do in the future.

  • The mechanical stability of the LbL is not checked for long range (shelf life),though in short term (14 days) is not good and cracked. This crack may be due to the hydrogen formation in localized pits in the substrate. The pH change is also the result of this process which prevented DNA retention.

Response: As indicated in the response above, the goal of the work was not exploring the long-term stability or shelf life of the system. The focus of the work was to demonstrate the efficacy of the dual treatment on creation of the LbL layers and the feasibility of the composite coatings to bind and release BSA, a mode protein for over two weeks. The SEM and AFM studies do not reveal any cracks in the film. The lines observed in the SEM image are not cracks but polishing lines from the SiC grit paper used to prepare the surface before generating the coatings. However, it is true that the PLGA is likely porous but there is no release of hydrogen since if this were the case, the pre-treated AZ31 would not show enhanced cell viability as shown in Fig. 6  and Fig. 7.

  • No mention of the cross section and thickness of the coating system is observed which is important on the integrity and stability of the coating in body fluid.

Response: It is extremely difficult to obtain a cross-sectional image with the polymer coating. Additionally, the coating thickness of pre-treated HF and Mg(OH)2 is well known in the literature. Furthermore, the focus of the current work was to explore and study the influence of alkali and HF pre-treatment on formation of LbL coatings. The EIS data and FTIR confirm the presence of the LbL films. Additionally, AFM also confirms the presence of the LbL coating and alteration of the surface roughness. A thicker coating also does not necessarily offer enhanced stability and integrity of the coatings. As we have shown, the dual alkali and acid pre-treatment renders the system resistant to hydrolysis by water. This is clear from the ICP measurements as well as the DNA quantification and Live/Dead assays. Hence, we believe that the system with the coatings as studied render them stable for the period of study of 10 days for BSA release and 21 days for dissolution.

  • The coating system shoed breakdown in Tafel polarization graph. EIS result does not show long term resistance of the coating and may mislead the workers.

Response: As mentioned above, the focus of the work was not to study the long-term stability and integrity of the films. The polarization curves and EIS data are only an indicator of the equilibrium response of the coatings. The current density is more reflective of the overall corrosion resistance. As is shown in Figure 10, the coatings clearly result in orders of magnitude decrease in corrosion current density showing the efficacy of the dual pre-treatment on the creation of the LbL layers and the resulting release of BSA. Long-term stability and shelf-life will be part of the future studies.   

Reviewer 3 Report

Multilayered films of PLGA and PAH were successfully synthesized using a straightforward LbL coating technique on AZ31 alloy substrates. The physicochemical characterization, surface morphologies, and microstructures of the LbL films were investigated, and the Electrochemical impedance spectroscopy measurements demonstrated that LbL multilayered coated substrates enhanced the corrosion resistance of bare MgAZ31 alloy. Cytocompatibility studies using human mesenchymal stem cells seeded directly over the substrates showed that the pretreated and LbL generated surfaces are more cytocompatible, displaying reduced cytotoxicity than the bare MgAZ31. The results are found to be interesting.

Comments:
1. A redundant "-1" appears in Figure 8a.

2. The corrosion current density (icorr) for AZ31, FHAZ31, and FHAZ31L was -5.098, 531 -6.509, and -6.163 log A/cm2, respectively. What is the corrosion current density in A/cm2?

3. In Figure 10b, what is the vertical axis of the Nyquist plots? There may be some problems.

4. The EIS was used to evaluate the corrosion resistance of the coating. The corrosion process needs to be further analyzed with the aid of equivalent circuit diagram.

Author Response

Answer to Reviewer 2. All corrections are highlighted in yellow.

  1. A redundant "-1" appears in Figure 8a.

Response: Thank you for pointing out this error. We have removed the redundant -1 from Figure 8a. The figure has been updated.

  1. The corrosion current density (icorr) for AZ31, FHAZ31, and FHAZ31L was -5.098, 531 -6.509, and

 -6.163 log A/cm2, respectively. What is the corrosion current density in A/cm2?

Response: We have adjusted the corrosion current density for AZ31, FHAZ31, and FHAZ31L to 7.97 x 10-6, 3.09 x 10-7 , and 6.87 x 10-7 A/cm2, respectively. These additions are incorporated in line 533-534 highlighted in yellow.

  1. In Figure 10b, what is the vertical axis of the Nyquist plots? There may be some problems.

Response: Thank you for pointing out this error. The vertical axis should be corrected to –Z’’(Ohms) and the horizontal axis should be corrected to Z’(Ohms). These corrections have been made in the updated Figure 10.

  1. The EIS was used to evaluate the corrosion resistance of the coating. The corrosion process needs to be further analyzed with the aid of equivalent circuit diagram.

Response: The EIS plots have been analyzed by fitting the experimental plot to the simulated scan generated by using the Zview software with the appropriate equivalent circuit. The equivalent circuit used for simulating each of the experimentally generated EIS plots are shown in the revised Figure 10 in the revised manuscript. The values obtained for each of the circuit parameters are also included in Table 1 in the revised manuscript. The corresponding text describing the changes in the circuit parameter values due to the different coating environment is also included in the text in lines 571-591 highlighted in yellow.

Reviewer 4 Report

1. Authors must follow the journal-style guidelines. For example, all e-mails of authors must be provided.
2. The strange alloy naming is used. What is MgAZ31 alloy? Is the substrate AZ31 commercial magnesium alloy or pure magnesium?
3. The meaning of abbreviations in the abstract must be provided.
4. Page 4, line 163: The authors used a JOEL microscope. Maybe JEOL?
5. Page 5, line 207: The word "medium" is repeated twice.
6. Fig. 2: As usual, the peaks on the XRD map are labeled by a phase name.

7. Page 8, line 329-330: The sentence "magnesium fluoride" is repeated twice.
8. Fig. 3: It is difficult to compare the bonds in the text with the peaks in the figure. Please label the bonds on the figure.
9. Page 9, lines 344-346: Is it sand paper? Please specify.
10. Why are cross-sections not provided? Why coating layer thickness is not determined?
11. Please compare the pH of HBSS medium with the initial pH of HBSS medium.

Author Response

All responses are highlighted in yellow.

  1. Authors must follow the journal-style guidelines. For example, all e-mails of authors must be provided.

Response:  We have added all authors and their email addresses in the revised submission highlighted in yellow.

  1. The strange alloy naming is used. What is MgAZ31 alloy? Is the substrate AZ31 commercial magnesium alloy or pure magnesium?

Response:  MgAZ31 is magnesium alloys containing aluminum. The Mg alloy used in this work has a chemical composition (in mass%) of 3% Al, 1% Zn. It is a commercially available magnesium alloy.

  1. The meaning of abbreviations in the abstract must be provided.

Response: The abbreviations for MgAZ31, FTIR, AFM, XRD, and SEM have been provided in the abstract highlighted in yellow. 

  1. Page 4, line 163: The authors used a JOEL microscope. Maybe JEOL?

Response: In page 4 line 172,the word JOEL is corrected as JEOL highlighted in yellow.

  1. Page 5, line 207: The word "medium" is repeated twice.

Response: In page 5 , line 216 the word repeated medium is removed.

  1. Fig. 2: As usual, the peaks on the XRD map are labeled by a phase name.

Response: The phase has been indicated by a label in Figure 2.

  1. Page 8, line 329-330: The sentence "magnesium fluoride" is repeated twice.

   Response: In page 8, line 338-339 the repeated word has been removed.

  1. Fig. 3: It is difficult to compare the bonds in the text with the peaks in the figure. Please label the bonds on the figure.

Response: The bonds have been labeled in Figure 3.

  1. Page 9, lines 344-346: Is it sand paper? Please specify.

Response: On page 9, line 353-354 the paper mentioned is different grit sized SiC polishing paper

  1. Why are cross-sections not provided? Why coating layer thickness is not determined?

Response: It is extremely difficult to obtain a cross-sectional image with the polymer coating. Additionally, the coating thickness of pre-treated HF and Mg(OH)2 is well known in the literature. Furthermore, the focus of the current work was to explore and study the influence of alkali and HF pre-treatment on formation of LbL coatings. The EIS data and FTIR confirm the presence of the LbL films. Additionally, AFM also confirms the presence of the LbL coating and alteration of the surface roughness.

  1. Please compare the pH of HBSS medium with the initial pH of HBSS medium

Response: The pH of the HBSS medium at 37oC is 7.3.  This was added in the text in line 512 of page 14.

Round 2

Reviewer 1 Report

The paper can be accepted without any further changes.